# IPA: An Information-Reconstructive Input Projection Framework for Efficient Foundation Model Adaptation

**Yuan Yin**                                              *yuan.yin@valeo.com*
*Valeo.ai, Paris, France*

**Shashanka Venkataramanan**              *shashanka.venkataramanan@valeo.com*
*Valeo.ai, Paris, France*

**Tuan-Hung Vu**                                    *tuan-hung.vu@valeo.com*
*Valeo.ai, Paris, France*

**Andrei Bursuc**                                    *andrei.bursuc@valeo.com*
*Valeo.ai, Paris, France*

**Matthieu Cord**                                    *matthieu.cord@valeo.com*
*Valeo.ai, Paris, France*
*Sorbonne Université, CNRS, ISIR, F-75005 Paris, France*

**Reviewed on OpenReview:** *https://openreview.net/forum?id=aLmQeZx2pR*

## Abstract

Parameter-efficient fine-tuning (PEFT) methods, such as LoRA, reduce adaptation cost by injecting low-rank updates into pretrained weights. However, LoRA's down-projection is randomly initialized and data-agnostic, discarding potentially useful information. Prior analyses show that this projection changes little during training, while the up-projection carries most of the adaptation, making the random input compression a performance bottleneck. We propose IPA, a feature-aware projection framework that explicitly aims to reconstruct the original input within a reduced hidden space. In the linear case, we instantiate IPA with algorithms approximating top principal components, enabling efficient projector pretraining with negligible inference overhead. Across language and vision benchmarks, IPA consistently improves over LoRA and DoRA, achieving on average 1.5 points higher accuracy on commonsense reasoning and 2.3 points on VTAB-1k, while matching full LoRA performance with roughly half the trainable parameters when the projection is frozen. Code available at https://github.com/valeoai/peft-ipa.

## 1 Introduction

Foundation models have transformed machine learning research and applications with their broad capabilities in language and visual understanding. These capabilities arise from model scale, since models with billions of parameters are pre-trained on massive corpora (Bommasani et al., 2021). However, adapting such general-purpose models to specialized tasks or domains remains challenging: as model sizes grow, full fine-tuning becomes computationally and financially expensive (Houlsby et al., 2019; Hu et al., 2022).

To address this bottleneck, the community has developed a range of parameter-efficient fine-tuning (PEFT) methods that reduce the number of trainable parameters by an order of magnitude compared to the base model (see surveys, e.g., Han et al., 2024; Zhang et al., 2025). Among these, Low-Rank Adaptation (LoRA; Hu et al., 2022) has gained significant traction due to its simplicity and effectiveness in the large-language-model community. In LoRA, each target weight matrix is reparameterized as the sum of the original pre-trained

weight $W$ and a low-rank update $\Delta W = BA$, where $A$ (the "down" projection) maps inputs into a lower-dimensional space and $B$ (the "up" projection) maps them back to the output dimension. This additional module can be merged into $W$ without introducing any extra latency during inference.

Although there has been a flurry of follow-up works around LoRA, most focus on alternative initializations (Meng et al., 2024; Yang et al., 2024) or extended structures (Liu et al., 2024; Huang et al., 2025; Albert et al., 2025) by restricting their analysis to the pretrained weight matrix, while paying little attention to the distribution of input features. In contrast, we broaden the focus to explicitly account for the role of input features.

In the original LoRA formulation, the down-projection matrix $A$ is randomly initialized and thus data-agnostic. Analyses of LoRA's inherent asymmetry show that during adaptation, this down-projection $A$ remains close to its initialization, whereas the up-projection $B$ adapts more effectively to the data (Tian et al., 2024; Hayou et al., 2024b; Zhu et al., 2024). This suggests that a data-agnostic input projection can become a performance bottleneck, motivating its replacement with a feature-aware, data-dependent alternative that better aligns with the intrinsic structure of the inputs.

In this paper, we tackle this direction and introduce IPA, an input-feature-aware projection scheme that preserves optimally the input information in the reduced hidden feature space. Our contributions are:

- We formulate adaptation with a dedicated feature-projection pretraining objective that favors information reconstruction in the bottleneck dimension through an encoder-decoder formulation.
- We instantiate this framework in the linear setting using efficient forward-only pretraining algorithms.
- We empirically validate IPA on language and vision-language tasks, showing consistent improvements over random linear projections. On several architectures, IPA matches the performance of fully trained LoRA while requiring roughly half as many trainable parameters.

## 2 Related Work

**Parameter efficient adapters.** PEFT techniques address the high computational cost of fine-tuning large foundational models by updating only a small set of parameters, rather than the full original network. A prominent class of PEFT methods is adapter-based: small trainable modules are added to a frozen model. Early work inserted bottleneck adapters between layers to enable task-specific tuning without altering original weights (Houlsby et al., 2019; Rebuffi et al., 2017); later designs placed adapters in parallel to existing layers for improved adaptation (He et al., 2022a). Recent work have explored structured parameterizations, e.g., Kronecker-factored matrices (Mahabadi et al., 2021). Li et al. (2024) employ block-specific adapter designs, dynamic parameter sharing, and mixtures of experts to improve efficiency and generalization. At the matrix level, LoRA and its variants constrain weight updates to a low-dimensional subspace for memory and compute-efficient tuning (Hu et al., 2022; Liu et al., 2024). Indeed, He et al. (2022a) show that many PEFT methods can be viewed through a unified lens of adapter.

Beyond architectural modifications, other PEFT strategies focus on minimizing the number of updated weights directly. These include sparse update methods (Guo et al., 2021; Sung et al., 2021; He et al., 2022b), which identify and tune only the most critical parameters. Recent work has even explored extremely low-precision adapters through quantization (Jie et al., 2023), demonstrating that 1-bit adapters can rival or surpass other PEFT strategies in both parameter efficiency and performance.

**LoRA methods and insights.** Among PEFT techniques, LoRA-based methods have emerged as particularly prominent due to their simplicity, inspiring a wide range of follow-up studies.

Several works aim to improve LoRA's design. Some focus on alternative initialization schemes. PiSSA (Meng et al., 2024) and CorDA (Yang et al., 2024) leverage spectral decompositions of the pretrained weights to initialize LoRA modules more effectively. Shuttleworth et al. (2024) observe that LoRA introduces novel singular directions absent in full fine-tuning. Building on this, LoRA-Null (Tang et al., 2025) initializes adapters in the nullspace of pretrained activations to reduce forgetting. Other approaches propose architectural modifications. DoRA (Liu et al., 2024) decomposes pretrained weights into basis and scaling components and applies LoRA on the basis. VeRA (Kopiczko et al., 2024) further simplifies this by fixing both $A$ and $B$ to

random bases and learning only scaling coefficients. RandLoRA (Albert et al., 2025) aggregates multiple VeRA-like components to achieve higher-rank updates. HiRA (Huang et al., 2025) follows a different route, applying element-wise multiplication between the LoRA module and the pretrained weight. These methods are all motivated by structural properties of the pretrained weights.

A parallel line of work investigates LoRA's learning behavior. Hayou et al. (2024b;a) analyze how imbalanced initialization affects feature-level dynamics during training. Zhu et al. (2024) report an asymmetry between the down- and up-projection matrices induced by standard initialization, which motivates subsequent variants such as HydraLoRA (Tian et al., 2024) and MALoRA (Wang et al., 2025). We refer the reader to Mao et al. (2025); Han et al. (2024) for more comprehensive overviews of LoRA and its many variants.

Our method differs from prior architectural improvements in that it also analyzes the input features to the target layers, rather than focusing solely on the pretrained weights. Drawing inspiration from studies on LoRA's learning behavior, our approach introduces a feature-aware projection objective that optimally preserves information in the input representation before applying the low-rank update.

## 3 IPA: Information-Reconstructive Input Projection for Adaptation

In this section, we begin by motivating the framework through empirical observations of standard LoRA training (Sections 3.1 and 3.2). Next, we present the IPA framework (Section 3.3), and finally provide a concrete instantiation (Section 3.4), which we evaluate in Section 4.

### 3.1 Preliminaries: LoRA

LoRA adapts a pretrained transformer-based model to a downstream task defined by a loss function $\mathcal{L}$ over a dataset $\mathcal{D}$. The pretrained model contains $K$ weight matrices denoted by $\{W^{(\ell)}\}_{\ell=1}^K$. For simplicity, we omit the layer index $\ell$ when not required. Each pretrained weight matrix $W \in \mathbb{R}^{d_{\text{out}} \times d_{\text{in}}}$ defines a linear map $f_W : x \in \mathbb{R}^{d_{\text{in}}} \mapsto z = Wx \in \mathbb{R}^{d_{\text{out}}}$, which transforms an input feature vector $x$ into an output feature vector $z$. Here, $d_{\text{in}}$ and $d_{\text{out}}$ denote the input and output dimensions of the layer, respectively.

To adapt a target layer, this linear map $f_W$ is augmented with two additional learnable maps: $f_A : x \in \mathbb{R}^{d_{\text{in}}} \mapsto x_h = Ax \in \mathbb{R}^r$, $f_B : x_h \in \mathbb{R}^r \mapsto Bx_h \in \mathbb{R}^{d_{\text{out}}}$, where $A \in \mathbb{R}^{r \times d_{\text{in}}}$ and $B \in \mathbb{R}^{d_{\text{out}} \times r}$ are low-rank matrices with a maximum rank $r \ll \max(d_{\text{in}}, d_{\text{out}})$. $x_h$ is the hidden feature projected by the mapping $f_A$. Adaptation proceeds over $T$ steps of gradient descent, with $t = 0$ denoting the initial state. We denote the matrices at step $t$ by $A_t$ and $B_t$. The modified forward pass of each target layer becomes

$$z = f_W(x) + \lambda f_{B_t}(f_{A_t}(x)) = Wx + \lambda B_t A_t x, \tag{1}$$

The elements of $A_0$ are drawn from a zero-mean Gaussian (or uniform) distribution and $B_0 = 0$, ensuring $W + B_0 A_0 = W$ while remaining trainable. The pretrained matrix $W$ remains frozen, and the positive scalar $\lambda$ rescales the low-rank residual update. In the original LoRA formulation, $\lambda = \frac{\alpha}{r}$, where $\alpha > 0$ is a tunable hyperparameter. Training LoRA thus implies computing gradients only for $A_t$ and $B_t$:

$$\nabla_{B_t}\mathcal{L} = \lambda \nabla_z \mathcal{L} x^\top A_t^\top, \qquad \nabla_{A_t}\mathcal{L} = \lambda B_t^\top \nabla_z \mathcal{L} x^\top, \tag{2}$$

leaving the pretrained weight unchanged.

### 3.2 Asymmetric Behaviors in LoRA

While LoRA has been widely adopted for efficient fine-tuning of large pretrained models, we observe a notable asymmetry between its two projection matrices: the down-projection matrix $A$ primarily serves to compress input features into a low-dimensional subspace, whereas the up-projection matrix $B$ plays the critical role of recombining those features to adapt the final model outputs. Notably, tuning $B$ alone while keeping $A$ fixed or randomly initialized often yields performance comparable to tuning both. This suggests that $B$ is mainly responsible for adapting the output, whereas $A$ serves as a feature projector.

To empirically illustrate this asymmetry, we conduct an adaptation experiment across multiple tasks. Following Huang et al. (2023), we choose the few-shot adaptation setting on the BIG-Bench Hard benchmark (BBH;

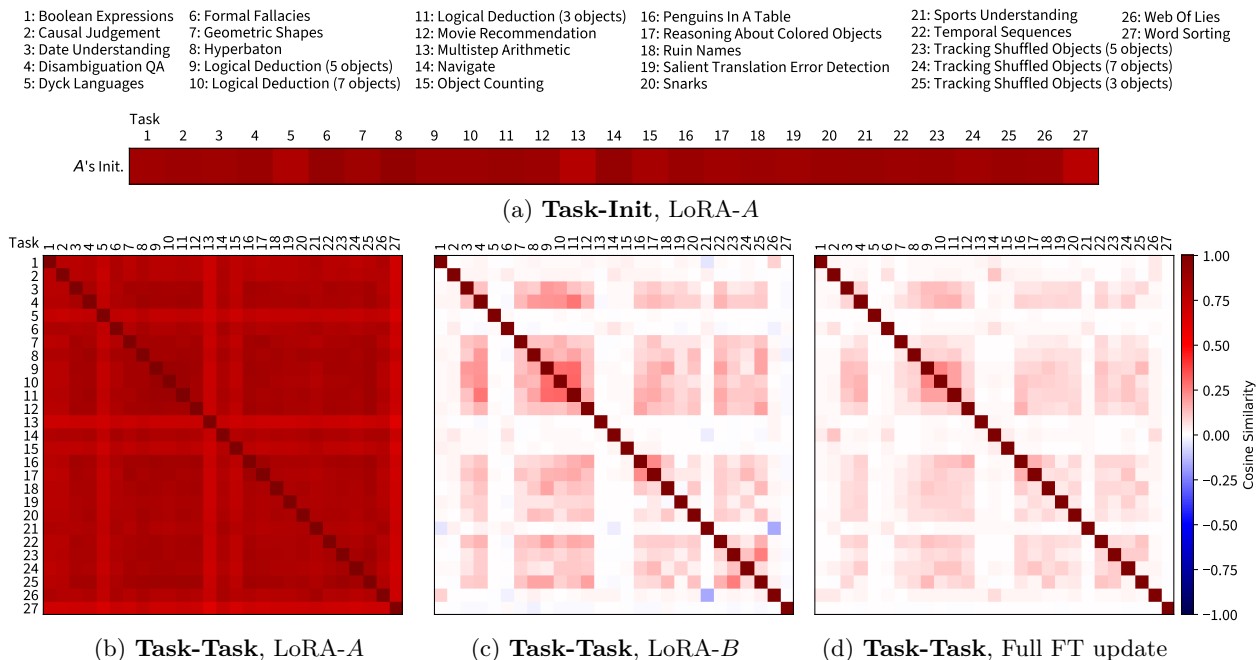

Figure 1: **Cosine-similarity matrices for LoRA and full-fine-tune updates on 27 BIG-Bench Hard tasks.** (a) shows the similarity between each trained LoRA-$A$ vector and its initialization; panels (b)–(d) show pairwise task–task similarities for LoRA-$A$, LoRA-$B$, and full fine-tune updates, respectively. The LoRA-$A$ vectors remain close to their shared initialization in (a) and vary little across tasks in (b), while the task-dependent patterns in LoRA-$B$ (c) closely match those from full fine-tuning (d).

Suzgun et al., 2023), which comprises 27 diverse tasks. We use Flan-T5 (Chung et al., 2024) as the base pretrained model. For each task $j$, we either fully fine-tune the pretrained model or learning LoRA adapters on a set of target layer $\Lambda$ for a fixed number of steps $T$, reaching zero training loss in both cases. All LoRA adapters are initialized with the same random seed across tasks, ensuring that $A_{0,j}^{(\ell)} = A_0^{(\ell)}$ for every target layer $\ell \in \Lambda$. This facilitates comparison of the learned LoRA matrices across tasks.

To analyze inter-task similarity, across all target layers $\Lambda$, we flatten and concatenate full-fine-tune updates and the trained LoRA matrices, yielding vectors for each task $j$: $\theta_{A,j} = \big\|_{\ell \in \Lambda} \text{vec}\big(A_{T,j}^{(\ell)}\big)$, $\theta_{B,j} = \big\|_{\ell \in \Lambda} \text{vec}\big(B_{T,j}^{(\ell)}\big)$, and $\Delta\theta_{W,j} = \big\|_{\ell \in \Lambda} \text{vec}\big(W_{T,j}^{(\ell)} - W^{(\ell)}\big)$. Figure 1 then presents cosine-similarity matrices for two cases: in panel (a) ("Task–Init, LoRA-$A$") we compare each trained vector $\theta_{A,j}$ to their common LoRA-$A$ initialization; panels (b)–(d) ("Task–Task, LoRA-$A$", LoRA-$B$ and Full FT, respectively) show pairwise similarities $\cos(\theta_{A,i}, \theta_{A,j})$, $\cos(\theta_{B,i}, \theta_{B,j})$, and $\cos(\Delta\theta_{W,i}, \Delta\theta_{W,j})$.

Remarkably, Figure 1a shows that $A$ matrices are still pretty similar to their initialization, while Figure 1b is largely uniform across tasks. This indicates that the learned $A$ matrices undergo little change during adaptation and capture minimal task-dependent variation. In contrast, Figures 1c and 1d reveal nearly identical block structures, suggesting that the task-specific information recovered by full fine-tuning is almost entirely absorbed by the $B$ matrices.

**Implications.** These findings indicate that the down-projection matrix $A$ in vanilla LoRA operates primarily as a random feature projector, rather than encodes the task-specific distinctions. Recent theoretical and empirical studies of LoRA (Hayou et al., 2024b; Zhu et al., 2024; Tian et al., 2024; Zhang et al., 2025) arrive at similar conclusions, showing that standard LoRA initialization induces pronounced asymmetries in both learning dynamics and representational behavior. Consequently, replacing this random feature projector with a more expressive, task-aware mapping could yield richer intermediate representations and improve adaptation performance.

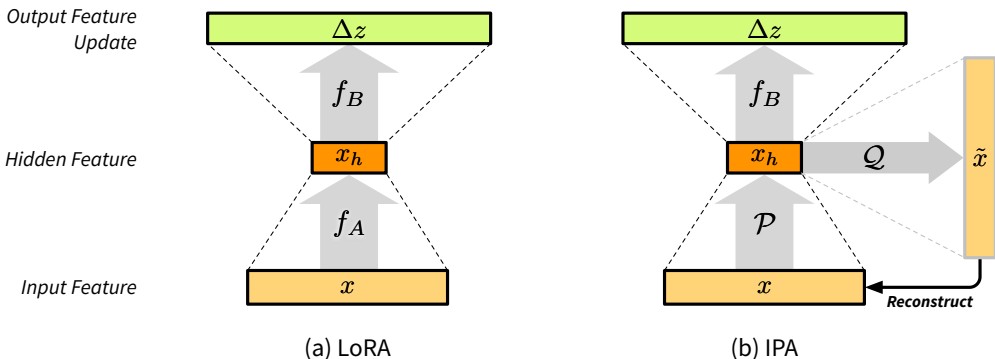

Figure 2: **Schematic of IPA vs. standard LoRA.** The gray arrows denote mappings between two vector feature spaces. In standard LoRA, an input feature $x$ is projected to a low-dimensional space by $f_A$ and then lifted back by $f_B$ to yield the update $\Delta z$. IPA introduces two pretrained projectors, $\mathcal{P}$ and $\mathcal{Q}$, enforcing that the hidden feature can reconstruct $x$; at adaptation time only $\mathcal{P}$ is retained.

### 3.3 The IPA Framework

We reinterpret the adaptation scheme by introducing a general feature projection function $\mathcal{P} \colon \mathbb{R}^{d_{\text{in}}} \to \mathbb{R}^{d_h}$ and write

$$z = f_W(x) + \lambda f_{B_t}(\mathcal{P}(x)) = Wx + \lambda B_t \mathcal{P}(x), \quad B_0 = 0.$$

where $\mathcal{P}$ maps the input $x$ into a hidden feature $x_h = \mathcal{P}(x) \in \mathbb{R}^{d_h}$. Therefore, full fine-tuning corresponds to $\mathcal{P}$ being the identity. It is thus natural to seek a closer approximation to full fine-tuning through more effective input compression.

**Information-reconstructive input projection.** When $d_h < d_{\text{in}}$, the projection $\mathcal{P}$ must compress $x$, which risks discarding task-relevant information if it is not chosen in an input-aware manner and the inputs lie in a subspace with larger intrinsic dimension. Standard LoRA initializes $\mathcal{P}$ as a random linear map, thus ignoring the input distribution. To address this, we instead seek $\mathcal{P}$ (and a complementary decoder $\mathcal{Q} \colon \mathbb{R}^{d_h} \to \mathbb{R}^{d_{\text{in}}}$) that minimize the reconstruction error:

$$\min_{\mathcal{P}, \mathcal{Q}} \mathbb{E}_{x \sim \mathrm{p}(x)} \|x - \tilde{x}\|^2, \quad \text{where} \quad \tilde{x} = \mathcal{Q}(\mathcal{P}(x)). \tag{3}$$

This objective encourages $\mathcal{P}$ to preserve as much information from the original input as possible, as measured by the $L^2$ reconstruction loss. Figure 2 contrasts IPA with standard LoRA, highlighting the additional optimization of the projector.

**Forward-only pretraining of projector.** Eq. (3) corresponds precisely to the objective of an autoencoder. One could therefore imagine training it with either linear or nonlinear functions for $\mathcal{P}$ and $\mathcal{Q}$. However, doing so for each modulated layer via backpropagation is impractical: the loss is difficult to integrate into the adapter training pipeline and incurs significant computational overhead compared to LoRA. Instead, we propose to learn the projector in a *forward-only* manner.

### 3.4 Instantiation: Linear Case

To instantiate the framework in practice, we must specify (i) the distribution of features used to pretrain the projector $\mathcal{P}$, (ii) the form of the projector $\mathcal{P}$, and (iii) the algorithm used to pretrain it.

**Pretraining distribution.** We pretrain $\mathcal{P}$ using target-domain hidden representations. Concretely, we pass training tokens through the frozen pretrained model and collect the resulting hidden states, forming a pretraining set $\hat{X} = [\hat{x}_i]_{i=1}^{N} \in \mathbb{R}^{N \times d_{\text{in}}}$. This ensures that $\hat{X}$ reflects both the model's internal feature and the target-domain data distributions.

**Projector architecture.**  To preserve LoRA's inference-time efficiency, we restrict $\mathcal{P}$ and its decoder $\mathcal{Q}$ to linear maps defined by a shared matrix $U \in \mathbb{R}^{d_h \times d_{\text{in}}}$:

$$\mathcal{P}(x) = Ux, \qquad \mathcal{Q}(x_h) = U^\top x_h.$$

In this setting, solving Eq. (3) is equivalent to performing PCA (Hotelling, 1933), by extracting the top-$d_h$ eigenvectors of the empirical covariance $\Sigma = \frac{1}{N}\hat{X}^\top \hat{X}$.

**Pretraining algorithm.**  Full PCA over all hidden states is infeasible due to storage and compute costs. Instead, we adopt incremental PCA (IPCA; Ross et al., 2008), which processes feature mini-batches sequentially and updates a low-rank approximation of $\Sigma$. Alternatives such as the generalized Hebbian algorithm (GHA; Sanger, 1989) also approximate principal components, but we found IPCA both more efficient and slightly more accurate in practice (see Section 4.4 for the ablation study).

**Default Configuration.**  Unless otherwise specified, we use target-domain hidden representations as input, a linear projector, and IPCA for pretraining. All main experiments adopt IPCA, and IPA refers to this implementation unless noted otherwise. The projector $U$ can optionally be refined by backpropagating the task loss, allowing it to adapt jointly with the residual weights and align more closely with the downstream objective. We analyze the effect of projector fine-tuning in Section 4.

## 4    Experiments

We evaluate IPA on both standard language- and vision-centric benchmarks, comparing it with leading PEFT baselines that rely on low-rank updates.

### 4.1    Experimental Setting

#### 4.1.1    Tasks, Datasets, Base Models, and Evaluation Protocol

**Language Tasks: Instruction Following**  We follow the adaptation protocol for instruction following proposed by Hu et al. (2023), where the adapted model must generate a response in the expected format from a set of given options. Each question is wrapped in a predefined natural language template with an instruction, and the training example is appended with the desired answer.

We use the `commonsense-170k` dataset, which contains 170,000 examples drawn from eight commonsense reasoning tasks. We study four recent strong pretrained large language models of moderate size: LLAMA-2 7B (Touvron et al., 2023), LLAMA-3 8B (Grattafiori et al., 2024), QWEN-2.5 7B (Qwen et al., 2024), and GEMMA-3 4B (Gemma Team et al., 2025). We only use the language model part if the model is multi-modal. All models are adapted once with the same dataset for 3 epochs and evaluated on the corresponding test splits for each reasoning task.

**Vision Tasks: Open-Vocabulary Image Classification**  We assess our method on open-vocabulary classification using the VTAB-1k benchmark (Zhai et al., 2019), which includes 19 tasks/datasets categorized into three groups: NATURAL, SPECIALIZED, and STRUCTURED. VTAB-1k uses exactly 1000 labeled examples per task for adaptation; the original test splits remain unchanged.[1]

We use the base variant of SIGLIP-2 (Tschannen et al., 2025) as the pretrained vision-language model. For each class, we construct a text embedding by inserting the class name into the prompt "`a photo of a [CLASSNAME]`" and encoding it with the model's text encoder. All embeddings are padded to the model's maximum sequence length. This prompt formulation is fixed across methods and orthogonal to the PEFT approach.

We adapt *only the vision encoder*, training it with a standard cross-entropy loss over logits computed with similarity scores between image and text embeddings. Logits are computed following the procedure used in

---

[1]We regenerated VTAB-1k using the code provided in Zhang et al. (2022), converting all examples to a lossless format instead of the lossy JPEG compression used in the original work. After this change, our zero-shot test performance for SIGLIP-2 matches the results reported in the model's original paper.

SIGLIP-2, for each image-class pair. Adaptation and evaluation are performed independently for each task. As in Zhang et al. (2022), we evaluate models every 10 epochs and report the best test performance observed over 100 training epochs.

### 4.1.2 Main Baselines

Our goal is to isolate and quantify the effect of the input projector introduced in IPA, w.r.t. random input projector-based methods. We therefore compare against two strong baselines from the same reparameterization PEFT family: • **LoRA** (Hu et al., 2022), which injects a low-rank residual update with a randomly initialized down-projection $f_A$. • **DoRA** (Liu et al., 2024), which decomposes each pretrained weight into a magnitude and direction: $W = \|W\| \cdot \frac{W}{\|W\|}$, where $\|W\|$ is the magnitude and $\frac{W}{\|W\|}$ is the unit-norm direction. Standard LoRA is then applied only to the direction, while the magnitude $\|W\|$ is fine-tuned jointly. For each method we evaluate two variants of the input projector: fixed (no gradient updates, denoted ✗) and trainable (fine-tuned during adaptation, denoted ✓), aligning with IPA's variants.

### 4.1.3 Hyperparameters

**Optimization.** To ensure a fair comparison across adaptation methods, we fixed the optimizer hyperparameters and learning rate scheduler in all experiments except the base learning rate. We employ Adam (Kingma & Ba, 2015) with a linear warm-up schedule: the learning rate is ramped up over the first steps, then decayed linearly. The base learning rate remains unchanged regardless of whether feature-projection fine-tuning is enabled. The same batch size is applied in each benchmark.

For each baseline, we align the recommended learning rates for baselines with those from Liu et al. (2024) for LLAMA-2 and LLAMA-3. We only deviate from these defaults when a more effective rate is identified. We choose a performing learning rate for more recent QWEN-2.5 7B and GEMMA-3 4B.

We report the detailed hyperparameters in Appendix A.

**Adapter configuration.** All methods share identical essential adapter settings to isolate differences coming solely from feature projection. Concretely, for language benchmark, we use identical hidden dimension $d_h = 32$ for the query, key, and value projections, as well as for the up/down projections in each MLP block, following Hu et al. (2023); Liu et al. (2024). For vision benchmarks, we restrict adapters to the query/value projections with $d_h = 8$, per standard practice in vision transformers as in Zhang et al. (2022). Therefore, the only methodological distinction lies in how feature projectors are trained. The scaling factor is fixed for all methods.

For instruction-following tasks, IPA uses a random 10% subset of the training data to construct the projector pretraining feature set. An ablation study examining the impact of this choice is presented in Figure 4b. For open-vocabulary classification, we leverage the full training set for each task, as each contains only 1000 examples.

## 4.2 Main Results

**IPA projector improves performance over random projection.** Tables 1 and 2 summarize our accuracy results on the instruction-following benchmark and the open-vocabulary classification tasks, respectively.

On the instruction-following benchmark at hidden dimension $d_h = 32$, IPA outperforms both LoRA and DoRA across most configurations and base models. For example, in Table 1, on LLAMA-3 8B without projector fine-tuning (25M trainable parameters, 0.31% of the model), IPA achieves an average accuracy of 85.6%, outperforming LoRA (85.0%) by 0.6 points and DoRA (84.7%) by 0.9 points. Even with projector fine-tuning (57M parameters, 0.70%), IPA still leads with 85.9%, compared to 85.5% for LoRA and 85.1% for DoRA. Similar gains are observed across other base models, yielding an average gain of 1.5 points.

On the open-vocabulary classification benchmark (Table 2), at hidden dimension $d_h = 8$, IPA reaches 73.7% group-level macro average accuracy without projector fine-tuning, surpassing LoRA by 3.0 points and DoRA by 2.8 points. With projector fine-tuning, performance improves to 76.5%, a 1.8-point gain over both baselines.

Table 1: **Comparison of instruction-following answer accuracy (%) on 8 commonsense reasoning benchmarks across multiple base models.** All methods are compared in the configuration with (✓) and without (✗) projector finetuning. We highlight the $\boxed{\text{best}}$ and the $\boxed{\text{second best}}$ scores under the same projector finetuning setting.

| Base model | Method | Proj. FT | Trainable Params (%) | BoolQ | PIQA | SocialIQA | HellaSwag | WinoGrande | ARC-easy | ARC-challenge | OpenbookQA | Avg. |
|---|---|---|---|---|---|---|---|---|---|---|---|---|
| Llama-2 7B | LoRA | ✗ | 28.0M (0.41%) | 60.5 | 78.7 | 74.5 | 76.3 | 75.1 | 82.8 | 66.1 | 76.8 | 73.8 |
| | DoRA | | 28.9M (0.43%) | 58.0 | 82.0 | 33.5 | 12.8 | 42.1 | 64.9 | 43.9 | 68.4 | 50.7 |
| | IPA (Ours) | | 28.0M (0.41%) | 71.7 | 83.2 | 80.0 | 89.0 | 82.0 | 84.8 | 70.1 | 79.0 | 80.0 |
| | LoRA | ✓ | 56.1M (0.83%) | 69.8 | 79.9 | 79.5 | 83.6 | 82.6 | 79.8 | 64.7 | 81.0 | 77.6 |
| | DoRA | | 57.0M (0.84%) | 71.8 | 83.7 | 76.0 | 89.1 | 82.6 | 83.7 | 68.2 | 82.4 | 79.7 |
| | IPA (Ours) | | 56.1M (0.83%) | 71.1 | 84.4 | 80.9 | 90.5 | 82.7 | 85.6 | 71.5 | 81.4 | 81.1 |
| Llama-3 8B | LoRA | ✗ | 25.2M (0.31%) | 73.6 | 88.1 | 80.3 | 95.0 | 85.2 | 90.4 | 80.1 | 87.4 | 85.0 |
| | DoRA | | 26.0M (0.32%) | 74.3 | 87.9 | 79.7 | 95.3 | 84.2 | 90.3 | 79.5 | 86.2 | 84.7 |
| | IPA (Ours) | | 25.2M (0.31%) | 74.8 | 88.6 | 81.1 | 95.4 | 85.6 | 91.7 | 79.9 | 87.8 | 85.6 |
| | LoRA | ✓ | 56.6M (0.70%) | 75.4 | 88.6 | 80.7 | 95.4 | 86.2 | 91.2 | 80.1 | 86.1 | 85.5 |
| | DoRA | | 57.4M (0.71%) | 75.3 | 89.3 | 80.8 | 95.3 | 85.8 | 89.9 | 79.3 | 85.6 | 85.1 |
| | IPA (Ours) | | 56.6M (0.70%) | 75.0 | 89.9 | 81.2 | 96.0 | 85.9 | 91.2 | 79.6 | 88.4 | 85.9 |
| Qwen-2.5 7B | LoRA | ✗ | 24.3M (0.32%) | 62.8 | 89.3 | 79.9 | 94.6 | 83.1 | 95.9 | 88.6 | 91.4 | 85.7 |
| | DoRA | | 25.1M (0.33%) | 62.0 | 89.8 | 78.6 | 94.6 | 83.0 | 96.1 | 88.9 | 89.8 | 85.3 |
| | IPA (Ours) | | 24.3M (0.32%) | 73.3 | 90.0 | 80.2 | 95.0 | 85.2 | 95.8 | 88.8 | 92.4 | 87.6 |
| | LoRA | ✓ | 54.1M (0.71%) | 63.5 | 89.8 | 79.5 | 95.4 | 85.9 | 95.9 | 88.3 | 92.2 | 86.3 |
| | DoRA | | 54.9M (0.72%) | 74.5 | 90.0 | 80.2 | 95.4 | 85.9 | 95.7 | 87.7 | 91.8 | 87.6 |
| | IPA (Ours) | | 54.1M (0.71%) | 74.5 | 90.0 | 79.7 | 95.3 | 85.5 | 96.2 | 88.7 | 92.0 | 87.7 |
| Gemma-3 4B | LoRA | ✗ | 21.4M (0.49%) | 69.3 | 84.4 | 78.2 | 90.6 | 80.3 | 89.5 | 76.4 | 82.0 | 81.3 |
| | DoRA | | 22.0M (0.51%) | 69.1 | 84.2 | 77.9 | 91.0 | 80.5 | 89.4 | 78.1 | 82.2 | 81.5 |
| | IPA (Ours) | | 21.4M (0.49%) | 68.7 | 85.0 | 78.5 | 90.0 | 81.5 | 90.3 | 78.0 | 84.4 | 82.0 |
| | LoRA | ✓ | 46.6M (1.07%) | 70.3 | 86.0 | 79.7 | 93.1 | 82.3 | 89.7 | 79.7 | 84.4 | 83.1 |
| | DoRA | | 47.3M (1.09%) | 70.6 | 85.3 | 80.0 | 92.9 | 82.8 | 90.0 | 77.6 | 85.4 | 83.1 |
| | IPA (Ours) | | 46.6M (1.07%) | 69.8 | 86.3 | 78.8 | 93.4 | 83.3 | 90.7 | 80.3 | 86.0 | 83.6 |

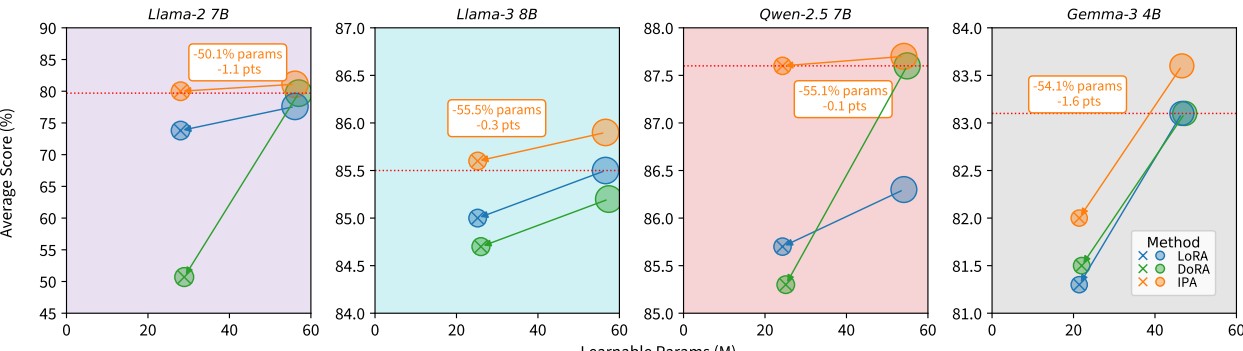

Figure 3: Comparison of IPA with baselines in both settings, with (◯) and without (⊗) finetunable feature projection on the commonsense benchmark. The dotted red line marks the highest baseline performance.

We also note that, across both benchmarks, LoRA remains competitive with DoRA, as the two methods yield similar scores in most settings when the learning rates are aligned.

**IPA suffers less from fixing input projectors.** The performance degradation from fixing the input projectors appears to be less pronounced for IPA in several cases. We showcase this in Figure 3. For instance, with Llama-2 7B, fixing the IPA projectors results in only a 1.1-point drop, compared to 3.8 points for

Table 2: **Top-1 accuracy (%) on 19 VTAB-1k open-vocabulary classification tasks with the SIGLIP-2 base model.** Methods are evaluated with (✓) or without (✗) projector finetuning. "Vision QV FT" finetunes the query/value projection layers of the vision encoder, while "Vision Full FT" finetunes all its parameters. We highlight the best and second best scores under the same setting. We report per-group averages ("G1/G2/G3 Avg."), the *Macro Avg.* (mean of group averages), and the *Micro Avg.* (mean over all tasks). *Significance:* for each setting (w/o and w/ projector finetuning), we run paired sign tests (two sided, $H_0\colon p = \frac{1}{2}$) comparing IPA against each baseline (LoRA, DoRA) over the 19 tasks. The "Sign $p$" column in a row reports the $p$-value for IPA vs. the method in that row (IPA rows use "—").

| Method | Proj. FT | Trainable Params (%) | Group 1: NATURAL | | | | | | | | Group 2: SPECIALIZED | | | | | Group 3: STRUCTURED | | | | | | | | | Macro Avg. | Micro Avg. | Sign $p$ |
|---|---|---|---|---|---|---|---|---|---|---|---|---|---|---|---|---|---|---|---|---|---|---|---|---|---|---|---|
| | | | Caltech101 | CIFAR-100 | DTD | Flowers102 | Pets | Sun397 | SVHN | G1 Avg. | Camelyon | EuroSAT | Resisc45 | Retinopathy | G2 Avg. | Clevr-Count | Clevr-Dist | DMLab | KITTI-Dist | dSpr-Loc | dSpr-Ori | sNORB-Azim | sNORB-Elev | G3 Avg. | | | |
| Zero-shot | | 0 (0.00%) | 84.4 | 73.9 | 63.0 | 84.1 | 94.9 | 61.2 | 28.6 | 70.0 | 50.9 | 40.0 | 62.8 | 5.0 | 39.7 | 27.9 | 20.0 | 17.0 | 4.2 | 6.4 | 4.8 | 5.2 | 10.4 | 12.0 | 40.5 | 36.7 | — |
| Vision QV FT | — | 14.2M ( 3.8%) | 94.2 | 78.3 | 80.2 | 98.2 | 93.3 | 66.6 | 93.2 | 85.2 | 85.3 | 96.5 | 91.0 | 74.8 | 86.9 | 85.0 | 60.2 | 48.5 | 85.1 | 88.2 | 52.2 | 37.2 | 43.6 | 62.5 | 78.2 | 75.5 | — |
| Vision Full FT | | 92.9M (24.8%) | 94.8 | 81.5 | 81.3 | 98.3 | 94.7 | 67.7 | 93.2 | 86.1 | 85.0 | 96.3 | 91.5 | 75.1 | 87.0 | 84.1 | 60.8 | 42.8 | 86.1 | 51.5 | 83.0 | 29.7 | 41.8 | 60.0 | 77.7 | 74.7 | — |
| LoRA | | 0.15M (0.039%) | 89.0 | 81.8 | 75.4 | 94.3 | 95.3 | 64.6 | 89.9 | 84.3 | 79.2 | 95.8 | 87.0 | 72.5 | 83.7 | 85.0 | 52.2 | 26.8 | 71.4 | 65.9 | 17.5 | 8.8 | 24.0 | 43.9 | 70.7 | 66.0 | 0.019 |
| DoRA | ✗ | 0.17M (0.044%) | 89.6 | 82.0 | 76.0 | 94.5 | 95.4 | 64.8 | 90.3 | 84.7 | 79.3 | 95.7 | 86.8 | 72.5 | 83.6 | 84.8 | 54.5 | 28.3 | 67.2 | 68.0 | 17.6 | 9.1 | 25.8 | 44.4 | 70.9 | 66.3 | 0.004 |
| IPA (Ours) | | 0.15M (0.039%) | 93.1 | 81.7 | 77.7 | 95.3 | 95.1 | 65.2 | 90.7 | 85.5 | 81.5 | 95.7 | 87.3 | 73.3 | 84.5 | 83.5 | 59.7 | 29.2 | 81.4 | 75.0 | 25.1 | 15.8 | 38.6 | 51.0 | 73.7 | 69.5 | — |
| LoRA | | 0.29M (0.079%) | 94.8 | 80.8 | 75.4 | 95.8 | 95.2 | 65.6 | 91.4 | 85.9 | 82.4 | 96.1 | 88.0 | 74.0 | 85.1 | 91.8 | 58.5 | 34.7 | 83.1 | 76.8 | 38.4 | 18.2 | 38.0 | 54.9 | 75.3 | 71.5 | 0.019 |
| DoRA | ✓ | 0.33M (0.083%) | 94.5 | 81.1 | 78.1 | 95.8 | 95.2 | 65.6 | 91.4 | 85.7 | 83.5 | 96.0 | 87.6 | 74.1 | 85.3 | 91.5 | 60.6 | 35.3 | 84.5 | 78.4 | 35.3 | 17.0 | 37.1 | 55.0 | 75.3 | 71.5 | 0.019 |
| IPA (Ours) | | 0.29M (0.079%) | 94.8 | 81.3 | 79.8 | 96.3 | 94.7 | 65.6 | 91.8 | 86.3 | 83.0 | 96.5 | 88.5 | 74.4 | 85.6 | 90.0 | 62.5 | 39.5 | 82.1 | 79.5 | 40.8 | 22.3 | 44.3 | 57.6 | 76.5 | 72.9 | — |

LoRA and a substantial 29.0 points for DoRA. On QWEN-2.5 7B, the drop is only 0.1 points for IPA, versus 0.6 and 2.3 points for LoRA and DoRA, respectively.

In Figure 3, the red dashed lines indicate the average accuracy of the best baseline with projector finetuning. We observe that IPA without projector finetuning matches or exceeds this baseline in 3 out of 4 models, while using less than half the tunable parameters. For instance, on LLAMA-2 7B, it requires 50.1% fewer parameters yet surpasses LoRA and DoRA by 3.5 and 0.3 points, respectively. On LLAMA-3 8B, it achieves comparable gains with 55.5% fewer parameters, outperforming LoRA and DoRA by 0.1 and 0.5 points.

These results further underscore the benefit of our input projector's information-preserving property.

## 4.3 Extended Comparison with Additional Baselines

We further broadened our evaluation on the commonsense reasoning benchmark using LLAMA-3 8B (cf. Section 4.1).

| Method | Avg. |
|---|---|
| PiSSA | 84.6 |
| OLoRA | 84.4 |
| CorDA | 80.8 |
| RandLoRA | 83.4 |
| VeRA ($r{=}512$) | 80.1 |
| LoRA | 85.5 |
| DoRA | 85.1 |
| IPA (Ours) | **85.9** |

Table 3: Comparison with additional PEFT methods.

| Projection | Proj. FT | Scaling $\lambda$ | Avg. |
|---|---|---|---|
| Random orthogonal | | 0.25 | 81.7 |
| Random orthogonal | ✗ | $0.25\sqrt{\frac{d_{\mathrm{in}}}{d_{\mathrm{h}}}}$ | 82.2 |
| LoRA | | 2 | 85.0 |
| IPA (Ours) | | 0.25 | **85.6** |
| Random orthogonal | | 0.25 | 85.5 |
| Random orthogonal | ✓ | $0.25\sqrt{\frac{d_{\mathrm{in}}}{d_{\mathrm{h}}}}$ | 83.1 |
| LoRA | | 2 | 85.5 |
| IPA (Ours) | | 0.25 | **85.9** |

Table 4: Comparison of projection types on commonsense.

**Comparison with additional baselines.** In Table 3, we evaluated IPA against a broader set of baselines, including methods that rely on random projections such as VeRA (Kopiczko et al., 2024) and RandLoRA (Albert et al., 2025), as well as approaches based on spectral or low-rank matrix decompositions such as PiSSA (Meng et al., 2024), OLoRA (Büyükakyüz, 2024), and CorDA (Yang et al., 2024). Within this

expanded set of techniques, IPA attains the highest average performance. All models use a hidden dimension $d_h = 32$, except VeRA. Additional per-task results and learning rate are provided in Table 6 in Appendix B.

**Comparison with random orthogonal projection.** In Table 4, we compared IPA with random orthogonal projections under matched hidden dimension ($d_h = 32$) and scaling configurations. Since random orthogonal projections reduce the norm of projected activations, we introduce an additional scaling factor to compensate for this effect and ensure a fair comparison with the norm-preserving PCA-based projection. In both scaling setups, random projection remains consistently below the performance of LoRA and IPA. See Appendix B, Table 7 for detailed results.

### 4.4 Ablation Studies

The following ablations also use LLAMA-3 8B on the instruction-following fine-tuning task.

**Projector pretraining algorithm.** As introduced in Section 3.4, we compare two online algorithms for estimating the top principal components: IPCA and GHA. Both optimize the same autoencoding objective Eq. (3). Table 5 reports results with and without projector fine-tuning. Across all settings, IPA-IPCA achieves higher downstream accuracy and converges more reliably than its GHA-based counterpart, making it our default choice.

Table 5: Comparison of instruction-following answer accuracy (%) between IPCA and GHA algorithms on commonsense reasoning benchmark.

| Method | Proj. FT | BoolQ | PIQA | SocialIQA | HellaSwag | WinoGrande | ARC-easy | ARC-challenge | OpenbookQA | Avg. |
|---|---|---|---|---|---|---|---|---|---|---|
| IPA-IPCA | ✗ | **74.8** | **88.6** | **81.1** | **95.4** | **85.6** | **91.7** | 79.9 | **87.8** | **85.6** |
| IPA-GHA | | 73.3 | 88.1 | 80.3 | 95.0 | 85.1 | 91.0 | **80.0** | 87.2 | 85.0 |
| IPA-IPCA | ✓ | **75.0** | **89.9** | 81.2 | **96.0** | 85.9 | **91.2** | 79.6 | **88.4** | **85.9** |
| IPA-GHA | | 74.9 | 89.3 | **81.3** | 95.8 | **86.3** | 90.4 | **80.1** | 86.2 | 85.6 |

**Projector pretraining set size.** The `commonsense-170k` dataset is large enough to investigate how the size of the projector pretraining set affects downstream performance. In Figure 4b, we pretrain the projector on randomly shuffled subsets ranging from 1% to 100% of the data, using a fixed seed for reproducibility. We select the first X% of examples from the shuffled split. Although performance generally improves up to around 10% of the data, we observe mitigated results beyond that point, which is likely due to variance in sample composition and/or randomized version of IPCA. Pretraining the feature projector on the full feature set takes roughly 1.7 hours on a NVIDIA H100 GPU, which is about ten times longer than using a 10% subset ($\approx$10 minutes). Note that adapter tuning on the full dataset requires about 5 hours for 3 epochs. Despite the substantially lower cost, the full dataset yields negligible or no accuracy improvement (and occasionally slight degradation due to variance), so we conclude that 10% is a practical sweet spot for efficient pretraining on `commonsense-170k` dataset without sacrificing downstream performance.

**Wall-clock time and memory consumption.** With the same setup as described above, pretraining the projector on a 10% subset adds only about 10 minutes to the total fine-tuning time of roughly 5 hours for 3 epochs on a single NVIDIA H100 GPU, resulting in a runtime comparable to LoRA. In contrast, DoRA requires around 10 hours under the same configuration. Peak memory usage is dominated by the fine-tuning phase and remains similar across methods (82 GiB for LoRA and IPA, 81 GiB for DoRA). To fairly compare IPA and LoRA with similar computational budget, we also compared IPA (with 10%-data pretraining) to LoRA trained for 3% more steps, yielding average scores of 85.9 and 85.0 respectively, showing that the small pretraining cost consistently translates into a performance gain. Full ablation results are given in Appendix B.

**Projected feature dimension.** In our ablation study, we vary the hidden dimension $d_h$ for IPA, LoRA, and DoRA, while keeping the learning rate, pretraining set size, and scaling ratio fixed. Figure 4a shows a characteristic bell-shaped curve for both IPA and LoRA: accuracy falls off steeply at very low dimensions, reaches a maximum over an intermediate range, then gradually declines as $d_h$ increases further. Importantly, IPA is more robust than LoRA: at $d_h = 8$, it matches LoRA's performance at $d_h = 16$, whereas LoRA's accuracy drops sharply. DoRA maintains a relatively flat performance profile across all tested dimensions but underperforms IPA once $d_h \geq 8$. For intermediate dimensions ($d_h = 16, 32, 64$), LoRA still outperforms DoRA. Detailed per-task results are provided in Appendix B.

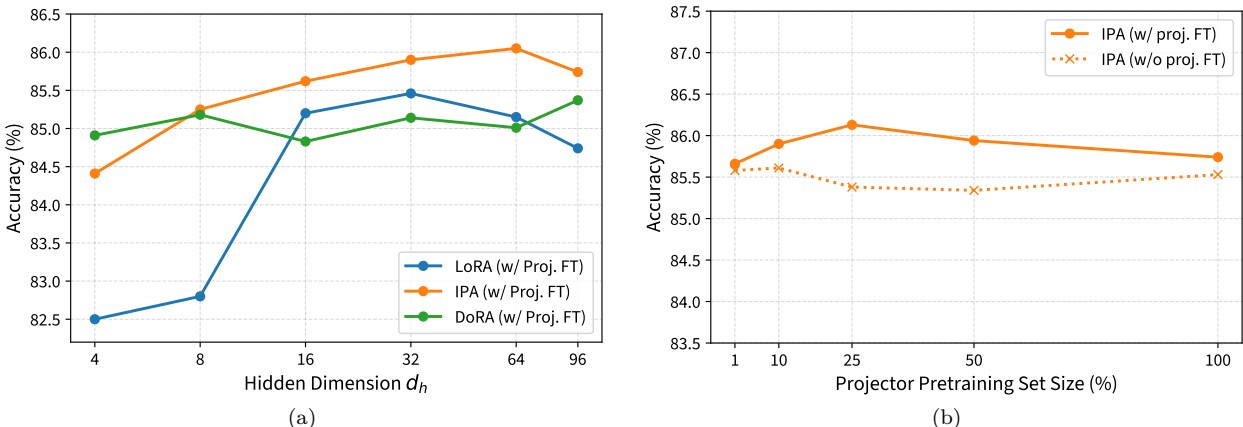

Figure 4: Average accuracy of LLAMA-3 8B models fine-tuned on commonsense benchmark with (a) varying hidden dimension $d_h$ for IPA, compared to LoRA and DoRA, both with input projection fine-tuning ● ● ●, and (b) IPA (with projection fine-tuning ● or without ×) with varying percentage of the training dataset to obtain the projection pretraining feature set.

## 5 Discussion and Conclusion

In this paper, we presented the IPA framework, which addresses parameter-efficient adaptation by focusing on improving input-feature projection. We augment the feature projector with an input-feature-aware optimization objective, allowing it to learn meaningful projections without requiring gradient backpropagation. In our instantiation, we employ a batched PCA algorithm for unsupervised feature projection, demonstrating that this simple approach can be trained efficiently and effectively.

Through experiments on both language and vision-language benchmarks, we showed that IPA consistently outperforms existing PEFT methods that rely on random feature projections when the setting is aligned. These results confirm that incorporating data-driven projections yields more expressive and adaptable models while maintaining low additional parameter cost.

Future work could investigate alternative projection techniques, novel optimization objectives for the projector and the incorporation of other backpropagation-free unsupervised or self-supervised learning methods. Such enhancements may further boost adaptation performance while retaining the computational efficiency of our framework.

## Acknowledgment

This work was granted access to the HPC resources of IDRIS under the allocation 2024-AD011015854 made by GENCI.

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

## A  Experimental Detail

**Hyperparameters.**  The hyperparameters used across all models are summarized as follows. For instruction-following tasks, we adopt a batch size of 16, aligning with Hu et al. (2023) and Liu et al. (2024). For open-vocabulary image classification, we use a batch size of 64.

We use a learning rate of $3 \times 10^{-4}$ for LLAMA-2 7B, and $1 \times 10^{-4}$ for LLAMA-3 8B, QWEN-2.5 7B, and GEMMA-3 4B. For all, LoRA and DoRA use a scaling factor ($\lambda = \frac{\alpha}{d_h}$) of 2, while IPA uses 0.25, except for GEMMA-3 4B, where it is 0.4. For SIGLIP 2, we apply a learning rate of $1 \times 10^{-3}$, scaling factors of 2 (LoRA/DoRA) and 0.5 (IPA), with a dropout rate of 0.1 across all variants.

## B  Additional Illustration and Experimental Results

**Feature-wise similarity.**  To complement the global similarity shown in Figure 1a through the cosine similarity of LoRA's $A$ matrices with respect to their initialization, we present in Figure 5 a comparison based on feature-wise similarity. We compute the expected cosine similarity of the projected features as follows: using the pretraining features $\hat{x}$ introduced in Section 3.4, we calculate, for each task $j$, the cosine similarity across the layers $\Lambda$ where LoRA is applied:

$$\frac{\sum_{\ell \in \Lambda} \sum_{\hat{x}^{(\ell)}} \left( (A_0^{(\ell)} \hat{x}^{(\ell)})^\top A_{T,j}^{(\ell)} \hat{x}^{(\ell)} \right)}{\sqrt{\sum_{\ell \in \Lambda} \sum_{\hat{x}^{(\ell)}} \left( (A_0^{(\ell)} \hat{x}^{(\ell)})^\top A_0^{(\ell)} \hat{x}^{(\ell)} \right)} \sqrt{\sum_{\ell \in \Lambda} \sum_{\hat{x}^{(\ell)}} \left( (A_{T,j}^{(\ell)} \hat{x}^{(\ell)})^\top A_{T,j}^{(\ell)} \hat{x}^{(\ell)} \right)}}.$$

On average, the projected features remain moderately similar to the original projected features across tasks.

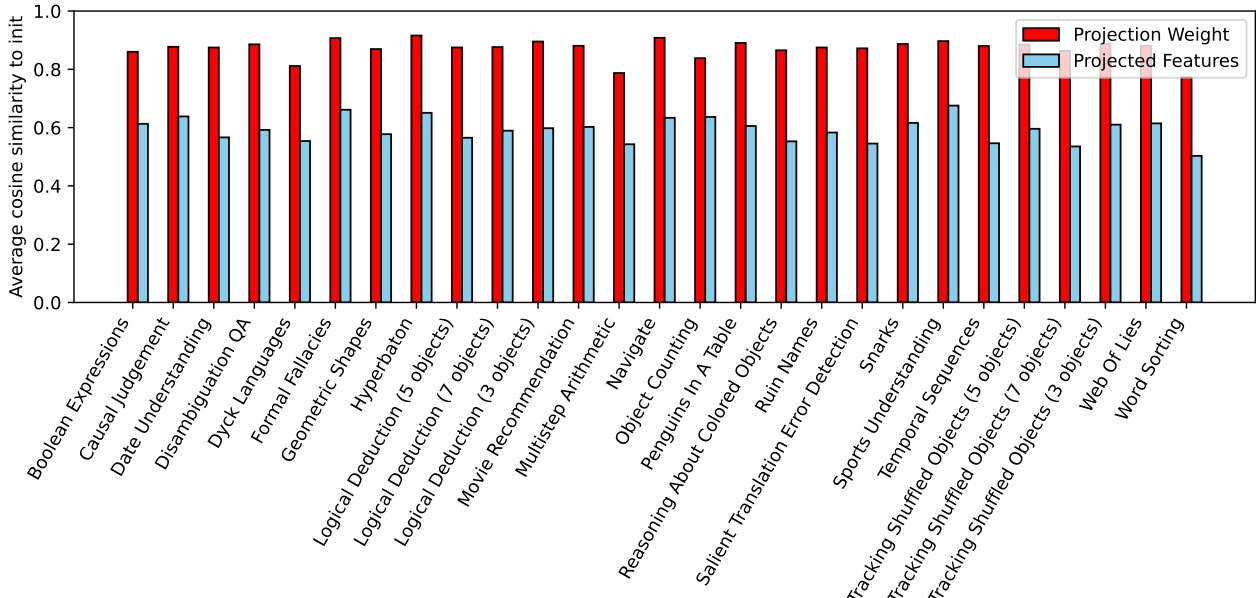

Figure 5: **Cosine similarity of LoRA $A$ projections after fine-tuning on 27 BBH tasks, compared to the initial projection, versus the similarity of projected features.** To capture the local behavior of the projection, we complement Figure 1a with a measure of feature-wise similarity between the projected representations.

**Detailed results of additional baselines.**  Table 6 reports the full per-task results for the extended set of baselines introduced in Table 3.

| Method | Learning Rate | BoolQ | PIQA | SocialIQA | HellaSwag | WinoGrande | ARC-easy | ARC-challenge | OpenbookQA | Avg. |
|---|---|---|---|---|---|---|---|---|---|---|
| LoRA | $1 \times 10^{-4}$ | 75.4 | 88.6 | 80.7 | 95.4 | 86.2 | 91.2 | 80.1 | 86.1 | 85.5 |
| DoRA | $1 \times 10^{-4}$ | 75.3 | 89.3 | 80.8 | 95.3 | 85.8 | 89.9 | 79.3 | 85.6 | 85.1 |
| PiSSA | $2 \times 10^{-5}$ | 75.2 | 88.6 | 81.1 | 94.8 | 85.5 | 89.8 | 77.1 | 84.6 | 84.6 |
| OLoRA | $2 \times 10^{-5}$ | 73.9 | 87.6 | 79.3 | 94.7 | 84.9 | 90.0 | 79.4 | 85.0 | 84.4 |
| CorDA | $2 \times 10^{-5}$ | 71.6 | 85.0 | 77.0 | 91.6 | 82.6 | 85.4 | 72.0 | 81.4 | 80.8 |
| RandLoRA | $1 \times 10^{-4}$ | 72.0 | 86.8 | 79.7 | 94.6 | 85.2 | 89.3 | 77.0 | 82.8 | 83.4 |
| VeRA ($r$=512) | $4 \times 10^{-3}$ | 62.2 | 85.2 | 77.8 | 91.7 | 80.3 | 87.2 | 75.2 | 81.6 | 80.1 |
| IPA | $1 \times 10^{-4}$ | 75.0 | 89.9 | 81.2 | 96.0 | 85.9 | 91.2 | 79.6 | 88.4 | 85.9 |

Table 6: Detailed results on comparison of with additional PEFT methods.

| Projection | Proj. FT | Scaling | BoolQ | PIQA | SocialIQA | HellaSwag | WinoGrande | ARC-easy | ARC-challenge | OpenbookQA | Avg. |
|---|---|---|---|---|---|---|---|---|---|---|---|
| Random ortho. | | 0.25 | 62.5 | 85.9 | 79.3 | 93.3 | 82.7 | 89.6 | 78.2 | 82.2 | 81.7 |
| Random ortho. | ✗ | $0.25\sqrt{\frac{d_{in}}{d_h}}$ | 65.9 | 86.9 | 79.5 | 94.1 | 84.1 | 89.1 | 75.8 | 81.6 | 82.2 |
| LoRA | | $\alpha/r = 2$ | 73.6 | 88.1 | 80.3 | 95.0 | 85.2 | 90.4 | 80.1 | 87.4 | 85.0 |
| IPA (Ours) | | 0.25 | 74.8 | 88.6 | 81.1 | 95.4 | 85.6 | 91.7 | 79.9 | 87.8 | 85.6 |
| Random ortho. | | 0.25 | 75.1 | 88.8 | 81.5 | 95.60 | 86.1 | 90.5 | 80.0 | 86.8 | 85.5 |
| Random ortho. | ✓ | $0.25\sqrt{\frac{d_{in}}{d_h}}$ | 63.3 | 87.3 | 80.5 | 94.0 | 85.2 | 90.0 | 78.8 | 85.8 | 83.1 |
| LoRA | | $\alpha/r = 2$ | 75.4 | 88.6 | 80.7 | 95.4 | 86.2 | 91.2 | 80.1 | 86.1 | 85.5 |
| IPA (Ours) | | 0.25 | 75.0 | 89.9 | 81.2 | 96.0 | 85.9 | 91.2 | 79.6 | 88.4 | 85.9 |

Table 7: Comparison of projection types on commonsense.

**Detailed results of ablation study.** Tables 8 and 9 show the detailed results of the ablation studies in Figures 4a and 4b in Section 4.4. Table 10 shows the detailed results on the ablation study about compute budget in Section 4.4.

Table 8: Detailed results of the ablation study on different hidden dimensions.

| Method | Proj. FT | Hidden Dim. | BoolQ | PIQA | SocialIQA | HellaSwag | WinoGrande | ARC-easy | ARC-challenge | OpenbookQA | Avg. |
|---|---|---|---|---|---|---|---|---|---|---|---|
| LoRA | ✓ | 4 | 62.1 | 87.9 | 78.9 | 91.3 | 84.0 | 89.9 | 79.4 | 86.6 | 82.5 |
| | | 8 | 62.1 | 88.8 | 80.5 | 92.3 | 83.0 | 90.2 | 80.7 | 84.8 | 82.8 |
| | | 16 | 74.7 | 87.4 | 80.9 | 95.4 | 86.7 | 90.0 | 79.4 | 87.2 | 85.2 |
| | | 32 | 75.4 | 88.6 | 80.7 | 95.4 | 86.2 | 91.2 | 80.1 | 86.1 | 85.5 |
| | | 64 | 75.1 | 88.4 | 81.0 | 93.0 | 86.9 | 90.4 | 79.7 | 86.8 | 85.1 |
| | | 96 | 74.9 | 88.4 | 79.8 | 94.6 | 86.3 | 89.6 | 78.8 | 85.4 | 84.7 |
| DoRA | ✓ | 4 | 73.6 | 88.6 | 79.8 | 95.5 | 85.1 | 90.2 | 80.3 | 86.2 | 84.9 |
| | | 8 | 75.6 | 89.1 | 80.7 | 95.6 | 85.2 | 90.9 | 78.7 | 85.8 | 85.2 |
| | | 16 | 73.5 | 88.9 | 80.2 | 95.3 | 86.1 | 90.5 | 78.6 | 85.6 | 84.8 |
| | | 32 | 75.3 | 89.3 | 80.8 | 95.3 | 85.8 | 89.9 | 79.3 | 85.6 | 85.1 |
| | | 64 | 74.8 | 88.6 | 80.9 | 94.9 | 85.3 | 89.4 | 79.9 | 86.2 | 85.0 |
| | | 96 | 74.6 | 89.0 | 80.0 | 95.3 | 85.9 | 90.4 | 79.0 | 88.8 | 85.4 |
| IPA | ✓ | 4 | 73.7 | 88.0 | 79.2 | 95.0 | 84.0 | 89.9 | 79.7 | 85.8 | 84.4 |
| | | 8 | 73.7 | 89.0 | 81.1 | 95.6 | 86.3 | 91.0 | 80.1 | 85.2 | 85.2 |
| | | 16 | 74.6 | 88.9 | 80.6 | 96.0 | 85.1 | 91.0 | 80.3 | 88.6 | 85.6 |
| | | 32 | 75.0 | 89.9 | 81.2 | 96.0 | 85.9 | 91.2 | 79.6 | 88.4 | 85.9 |
| | | 64 | 75.9 | 88.4 | 80.4 | 95.9 | 87.5 | 91.5 | 81.0 | 87.8 | 86.1 |
| | | 96 | 75.6 | 88.2 | 81.4 | 95.9 | 86.6 | 91.0 | 80.5 | 86.8 | 85.7 |

Table 9: Detailed results of the ablation study on projector pretraining set size.

| Method | Proj. FT | Proj. Pre-training Set | BoolQ | PIQA | SocialIQA | HellaSwag | WinoGrande | ARC-easy | ARC-challenge | OpenbookQA | Avg. |
|---|---|---|---|---|---|---|---|---|---|---|---|
| IPA | ✗ | 1% | 74.1 | 88.5 | 80.9 | 95.3 | 86.1 | 91.4 | 80.8 | 87.6 | 85.6 |
| | | 10% | 74.9 | 88.5 | 81.0 | 95.7 | 85.6 | 91.0 | 80.0 | 88.2 | 85.6 |
| | | 25% | 73.6 | 88.2 | 80.5 | 95.5 | 85.8 | 91.0 | 80.1 | 88.4 | 85.4 |
| | | 50% | 74.3 | 88.2 | 80.7 | 95.3 | 85.4 | 90.2 | 80.4 | 88.2 | 85.3 |
| | | 100% | 73.7 | 88.0 | 81.1 | 95.2 | 86.6 | 90.7 | 80.1 | 88.8 | 85.5 |
| | ✓ | 1% | 75.2 | 88.8 | 81.0 | 95.6 | 86.5 | 91.3 | 79.6 | 87.2 | 85.7 |
| | | 10% | 75.0 | 89.9 | 81.2 | 96.0 | 85.9 | 91.2 | 79.6 | 88.4 | 85.9 |
| | | 25% | 75.4 | 89.4 | 81.8 | 96.0 | 88.1 | 91.1 | 79.9 | 87.4 | 86.1 |
| | | 50% | 74.9 | 89.2 | 81.5 | 95.9 | 87.6 | 91.1 | 80.7 | 86.6 | 85.9 |
| | | 100% | 75.1 | 88.8 | 80.8 | 96.1 | 86.9 | 90.9 | 79.9 | 87.6 | 85.7 |

Table 10: Comparison between LoRA and IPA under matched computational budgets. LoRA is trained for about 3% more steps to match total training time of IPA pretrained with 10% of training data.

| Method | Proj. FT | FT Steps | BoolQ | PIQA | SocialIQA | HellaSwag | WinoGrande | ARC-easy | ARC-challenge | OpenbookQA | Avg. |
|---|---|---|---|---|---|---|---|---|---|---|---|
| LoRA | ✓ | 32,887 | 76.2 | 88.0 | 80.1 | 95.0 | 86.7 | 90.6 | 78.5 | 84.4 | 85.0 |
| IPA | ✓ | 31,926 | 75.0 | 89.9 | 81.2 | 96.0 | 85.9 | 91.2 | 79.6 | 88.4 | 85.9 |

