# OpenReview forum: "IPA: An Information-Reconstructive Input Projection Framework for Efficient Foundation Model Adaptation"
_TMLR — Accepted by TMLR_

### Review · Reviewer_rN61 · 2025-09-25

**Summary Of Contributions:**

The authors pose a problem with the random-projection behavior of the LoRA down-projection weight matrix. They propose an alternative down-projection matrix construction by pretraining the projector matrix on the reconstruction task with downstream data. Empirical validation results show that the proposed framework somehow improves the alignment of large language models on the instruction following tasks.

# Strengths
- The authors deal with an important yet underexplored problem, i.e., the problem of better initialization for LoRA.
- The proposed framework is conceptually simple and makes sense.

# Weaknesses
- Lack of rigorous evidence and justification on the key claim (See the next sections).
- Insignificant performance improvements compared to the baseline.
- Limited volume of empirical studies in terms of baseline methods.

**Audience:**

Yes

**Audience Explanation:**

The authors nicely pose a phenomenon -- asymmetric weight learning -- in LoRA, and show how enhanced weight initialization brings better downstream adaptation. This observation offers plenty of insights, and some audiences in TMLR may enjoy this insight.

**Broader Impact Concerns:**

No concerns on the ethical side.

**Claims And Evidence:**

No

**Claims Explanation:**

Lack of justification for the claim "data-dependent projection would be better than the random projection for downstream adaptation".
  * There is much evidence that random projection quite well preserves the information of the original high-dimensional vector [1, 2, 3].
  * Given this knowledge, the authors' major claim on the effectiveness of data-dependent initialization over random initialization should be rigorously validated.
  * While the authors provide empirical results by comparing their method with LoRA, it is still unclear whether some of the performance gains are truly due to better initialization or just driven by longer training with more computation.




> Reference
1. Random projection in dimensionality reduction: Applications to image and text data, Bingham and Mannila 2001
2. MEASURING THE INTRINSIC DIMENSION OF OBJECTIVE LANDSCAPES, Li et al. 2018
3. Black-Box Tuning for Language-Model-as-a-Service, Sun et al. 2022

**Requested Changes:**

## critical to securing my recommendation for acceptance
- Please add more explicit evidence on whether the data-dependent projection is genuinely better than random projection.
- Please add a controlled experiment on LoRA with an extended computation budget, so that a fair comparison with the proposed IPA is possible.
- Please add more discussion on the trade-off between the computational overhead and performance gain of IPA compared to the vanilla LoRA.

## simply strengthen the work in my view
- On page 6, there is a little jump in the 'pretraining algorithm' paragraph.
  - The authors directly introduce the principal component analysis (PCA) as a learning objective of the proposed projector.
  - Adding a brief cushion sentence with a reference that explicitly mentions the equivalence between the linear autoencoder reconstruction learning with paired weight and the PCA would be helpful for broader readers who may be less familiar with this.
- Although there exist tons of LoRA variants, the authors only provide DoRA as a non-LoRA baseline. Could the authors add more baseline to enhance the completeness of this work?

---

> ### Author Response · Authors · 2025-10-14
>
> We thank the reviewer for the insightful remarks that helped refine our manuscript and strengthen the empirical analysis and comparison.
> ## On information preservation
> Thank you for pointing this out. We agree that random projection also preserves the input information to some degree, and we did not intend to claim that it is non-information-preserving. Rather, it is suboptimal in terms of reconstruction, which may lead to less effective downstream adaptation when compared side by side with full fine-tuning, where the zero-initialized model updates directly interact with the original features. Our goal is for the inputs to these zero-initialized updates to retain as much information from the original input as possible. While random projections primarily preserve pairwise distances (i.e., geometric relationships), IPA, based on PCA projections, focuses on optimally preserving feature information through reconstruction.
>
> We have revised the manuscript to emphasize reconstruction and to include a more explicit comparison with random projection in the following items.
> ## Explicit comparision to random projection
> We performed an ablation replacing IPA’s input projection with a random orthogonal one, with or without scaling by $\sqrt{d_\text{in}/d_\text{h}}$. The additional scaling ensures that the norm of the projected features remains comparable to that of the features projected onto the principal components.
> This experiment was conducted on the commonsense benchmark using LLaMA3-8B with a learning rate of $10^{-4}$ and a hidden dimension of 32.
>
> |Projection|Proj. FT|Scaling|Commonsense Avg.|
> |---|---|---|---|
> |Random orthogonal|No|0.25|81.71|
> |Random orthogonal|No|$0.25\times\sqrt{\frac{d_\text{in}}{d_\text{h}}}$|82.15|
> |LoRA|No|$\alpha/r=2$|85.00|
> |IPA-IPCA|No|0.25|85.64|
> |Random orthogonal|Yes|0.25|85.50|
> |Random orthogonal|Yes|$0.25\times\sqrt{\frac{d_\text{in}}{d_\text{h}}}$|83.08|
> |LoRA|Yes|$\alpha/r=2$|85.46|
> |IPA-IPCA|Yes|0.25|85.90|
>
> We observe that random orthogonal projections underperform IPA across both projection fine-tuning settings and are less robust than both IPA and LoRA when projection fine-tuning is disabled. We will include these results in an appropriate section.
> ## Wall-clock time and ablation study on LoRA with extensive budget
> For the commonsense benchmark, pretraining the projector on 10\% of the data ($\sim$17k examples) in IPA adds only about 10 minutes of overhead compared to the $\sim$5 hours required for 3 fine-tuning epochs shared by LoRA and IPA. In contrast, DoRA is noticeably slower, requiring around 10 hours under the same configuration. All timings were measured on a single NVIDIA H100 GPU on the same idle node to ensure fair comparison.
>
> We also conducted an ablation study under a matched compute budget, extending LoRA fine-tuning by $\sim$3\% more steps to align total wall-clock time with IPA.
>
> |Method|FT Steps|BoolQ|PIQA|SIQA|HellaS|WinoG|ARC-e|ARC-c|OBQA|Avg|
> |:-------|:---------:|:------:|:----:|:----:|:------:|:------:|:------:|:------:|:----:|:----:|
> |LoRA|32990|76.18|88.03|80.14|95.01|86.74|90.57|78.50|84.40|84.95|
> |IPA|31926|74.98|89.93|81.22|96.00|85.87|91.20|79.60|88.40|85.90|
>
> Even with an aligned compute budget, IPA achieves a higher average performance than LoRA. LoRA appears to show mild signs of overfitting in this setting. A dedicated paragraph has been added to the manuscript to clarify these details in the ablation study.
> ## Citation on PCA
> We added a citation to the classical result showing that the PCA basis provides the optimal linear projection that minimizes the reconstruction error.
> ## Stronger baselines
> We extended the baselines to broaden the scope of comparison, which was originally focused on the projection component. We compared IPA with more LoRA variants mentioned by Reviewer 4uE7 for LLaMA3-8B on the commonsense benchmark. Each method follows the recommended learning rate for instruction-following tasks, with the rank fixed to 32 for all methods except VeRA, which uses its default configuration.
>
> Results reproduced with HuggingFace's PEFT implementation:
>
> |Method|LR|BoolQ|PIQA|SIQA|HellaS|WinoG|ARC-e|ARC-c|OBQA|Avg|
> |--------|-------|-------|------|------|--------|-------|-------|-------|------|-----|
> |LoRA|1e-4|75.44|88.57|80.71|95.37|86.18|91.20|80.10|86.10|85.46|
> |DoRA|1e-4|75.26|89.30|80.76|95.32|85.79|89.86|79.27|85.60|85.14|
> |PiSSA|2e-5|75.20|88.63|81.12|94.83|85.48|89.77|77.13|84.60|84.59|
> |OLoRA|2e-5|73.91|87.60|79.32|94.74|84.85|89.98|79.44|85.00|84.36|
> |CorDA|2e-5|71.56|85.04|76.97|91.58|82.56|85.35|72.01|81.40|80.81|
> |RandLoRA|1e-4|71.96|86.83|79.68|94.64|85.24|89.31|77.05|82.80|83.44|
> |VeRA ($r=512$)|4e-3|62.17|85.15|77.79|91.66|80.27|87.21|75.17|81.60|80.13|
> |IPA|1e-4|74.98|89.93|81.22|96.00|85.87|91.20|79.60|88.40|85.90|
>
> As shown, IPA maintains its advantage over the baselines. We also plan to report LoRA-Null results once the model is fully operational and add the results to the manuscript.

---

> > ### Comment · Reviewer_rN61 · 2025-10-18
> >
> > I appreciate the authors' efforts to strengthen the supporting evidence for the proposed method as well as the paper writing. I think my concerns were all addressed somehow. But one last question is, can authors provide any further discussion on why IPA performs better on some datasets than LoRA, but underforms on others? Deeper discussion on this will make the insights much richer, so that audiences can enjoy plenty of takeaways.

---

### Review · Reviewer_VdwZ · 2025-10-02

**Summary Of Contributions:**

In this work the authors seek to improve the LORA finetuning method which updates weight matrices with a low-rank factorized version. They make the observation that in the LORA decomposition $W = W_{0}+BA$, the matrix $A$ often does not change much from its initialized value. They hypothesize that the initial value of $A$, representing a random feature map, may not be the best space in which to make changes.

The authors then propose IPA, a method which uses iterative SVD to choose an initialization for $A$ which preserves as much input information as possible. They then provide evidence on a variety of finetuning settings that IPA outperforms LORA and DORA more often than not, and that IPA suffers less degradation than the other methods when $A$ is frozen. They suggest that methods like IPA which more faithfully preserve information in the inputs are a good feature to have for low rank finetuning methods, that don't require much compute overhead.

**Audience:**

Yes

**Audience Explanation:**

Finetuning large models is an area of interest to the community. This work provides an clear explanation about the key phenomenology motivating their approach, which some readers may find helpful. The method itself seems interesting and promising which will also be of interest to the readership.

**Claims And Evidence:**

Yes

**Claims Explanation:**

The paper clearly presents the background of finetuning, and motivates the approach well using Figure 1. The proposed algorithm is well motivated, easy to implement, and easy to run. The experiments to the best of my knowledge seem compelling. I am not a subject matter expert in finetuning, so I will defer to the other reviewers on questions of appropriate datasets and tasks.

To the best of my knowledge this approach is new; but again I will defer to reviewers more versed in this area on this point.

**Requested Changes:**

The main methodological issue I have is that, if I understand Appendix A.1 correctly, the same learning rate was used for all methods. It's not clear to me that this is the best thing to do; for example, if the chosen learning rate is better adapted to IPA than to LORA, LORA may perform worse while actually not being worse. I ask that the authors provide experimental evidence that tuning the learning rate better for each method does not destroy the gains of IPA. I think this can be done on a subset of the tasks (in the setting with feature map finetuning) to be convincing. Adding such an experiment would be crucial to secure my recommendation for acceptance.

The remaining are comments about presentation.

Figure 1 took a while for me to parse; in particular I almost missed figure 1 (a) because of its size relative to the other figures. Is there some alternative way to display that information? I also would love the equivalent of 1 (a) but for B.

On the topic of Figure 1: something that was not clear to me is how much one should care about the absolute change in A, versus the change in the projections $Ax$ for vectors $x$ coming into the layer. Is there a way to display information on this? Something like:

$$C(0, t) = \frac{\mathbb{E}[x^{\top}A^{\top}\_{t} A\_{0} x]}{\sqrt{\mathbb{E}[x^{\top}A^{\top}\_{0} A\_{0} x]}\sqrt{\mathbb{E}[x^{\top}A^{\top}\_{t} A\_{t} x]}}$$

This more directly ties into the motivation of the approach as well.

For the table in Figure 2, it would be helpful to report what fraction of times IPA is the best (separately for projection FT/no projection FT), and maybe a basic statistical test showing that IPA wins more than expected by random chance (using either a Sign Test or Wilcoxon signed rank test, with null hypothesis of success $p=1/3$, or $p=1/2$ to be more stringent if you believe that LORA and DORA are very similar).

---

> ### Author Response · Authors · 2025-10-14
>
> We thank the reviewer for the helpful comments, especially regarding hyperparameter sensitivity and presentation clarity.
>
> ## Learning rate
>
> We followed standard practice for the commonsense benchmark, where a base learning rate of $10^{-4}$ is typically used.
> Nevertheless, we agree that a more extensive learning-rate sweep could offer a clearer view of relative robustness. The ablation below (hidden dimension $d_h=32$ with projection fine-tuning with LLaMA3-8B) confirms IPA's stability across learning rates:
>
> |Method|learning rate = $5\times 10^{-5}$|$1\times 10^{-4}$|$2\times 10^{-4}$|$3\times 10^{-4}$|
> |---|---:|---|---|---|
> |LoRA |85.50|85.46|83.44|80.79\*|
> |DoRA | 85.48|85.14\*|83.50|81.36|
> |IPA|85.23|85.90|85.70|84.06|
>
> (\*) corresponds to the learning rates reported in the DoRA paper, reproducing similar scores (as originally reported 80.5 for LoRA, 85.2 for DoRA). IPA remains robust, while LoRA and DoRA plateau around $10^{-4}$.
>
> ## Presentation
>
> ### Figure 1
>
> - We repositioned Figure 1a for improved readability.
> - A version for the $B$ matrices is omitted because its zero initialization renders cosine similarity undefined (0/0).
> - We also added Figure 5 in the appendix to complement Figure 1a. It shows feature-wise cosine similarity between the projected features from the initial and trained projection matrices as proposed by the reviewer, providing a more data-centric perspective.
>
> ### Table 2: Statistical test
> We followed the reviewer's suggestion and added paired sign tests across tasks to report statistical significance in the updated manuscript.

---

> > ### Comment · Reviewer_VdwZ · 2025-10-14
> > **Thank you for the updates**
> >
> > The improvements to the paper address my concerns, no further comments at this time.

---

### Review · Reviewer_4uE7 · 2025-10-04

**Summary Of Contributions:**

The paper argues that LoRA’s down‑projection (A) behaves largely as a random compressor while the up‑projection (B) carries the learning signal. It proposes IPA, which replaces A with a feature‑aware projector trained to preserve information (minimize reconstruction error) using forward‑only incremental PCA over hidden states from the target domain; only the encoder P is kept for adaptation (decoder Q is discarded). Across commonsense instruction‑following (Llama‑2/3, Qwen‑2.5, Gemma‑3), IPA improves over LoRA/DoRA and, when projectors are frozen, reaches LoRA‑level accuracy with roughly half the trainable parameters.

**Audience:**

Yes

**Audience Explanation:**

Parameter efficient fine-tuning is of great interest to the foundation model community.

**Claims And Evidence:**

Yes

**Claims Explanation:**

1. Compelling diagnosis of LoRA asymmetry. The cosine‑similarity study over 27 BBH tasks shows LoRA‑A stays near initialization while LoRA‑B mirrors full‑FT structure (Figure 1), crisply motivating an input‑aware projector。

2. Simple, practical formulation. IPA reduces to a linear autoencoder objective; with IPCA the projector is learned via forward passes only (Sec. 3.4; Fig. 2), preserving LoRA’s inference cost.

3. Consistent empirical gains and thoughtful ablations.

**Requested Changes:**

1. stronger baselines. Include PiSSA/CorDA (spectral init), VeRA/RandLoRA (fixed/random bases), and LoRA‑Null; even a subset of tasks would calibrate IPA against the most relevant contenders.

2. Limited theory. The paper motivates PCA empirically but offers no analysis linking PCA’s MSE optimality to downstream task risk; even a linear‑Gaussian argument would strengthen claims.

3. Scope of projection learning. Only linear projectors are tested. What about mild-linear projectors (e.g., tied two‑layer linear + activation)?

4. Generalization and reuse. It remains unclear how projectors transfer across tasks or share across layers/blocks; the paper notes optional fine‑tuning of P but does not study cross‑task robustness or projector sharing. Of course, it could be beyond scopes of this paper.

5. Compute and memory accounting. While pretraining time for IPCA is sketched, a fuller wall‑clock and memory profile vs. LoRA/DoRA (pretraining + tuning) would make the efficiency claim more concrete.

---

> ### Author Response · Authors · 2025-10-14
>
> We thank the reviewer for the constructive feedback, which helped improve the scope and completeness of our experiments.
>
> ## Stronger baselines
> We extended the baselines to broaden the scope of comparison, which was originally focused on the projection component. We compared IPA with more LoRA variants mentioned by the reviewer for LLaMA3-8B on the commonsense benchmark. Each method follows the recommended learning rate for instruction-following tasks, with the rank fixed to 32 for all methods except VeRA, which uses its default configuration.
>
> Results reproduced with HuggingFace's PEFT implementation:
>
> |Method|LR|BoolQ|PIQA|SIQA|HellaS|WinoG|ARC-e|ARC-c|OBQA|Avg|
> |--------|-------|-------|------|------|--------|-------|-------|-------|------|-----|
> |LoRA|1e-4|75.44|88.57|80.71|95.37|86.18|91.20|80.10|86.10|85.46|
> |DoRA|1e-4|75.26|89.30|80.76|95.32|85.79|89.86|79.27|85.60|85.14|
> |PiSSA|2e-5|75.20|88.63|81.12|94.83|85.48|89.77|77.13|84.60|84.59|
> |OLoRA|2e-5|73.91|87.60|79.32|94.74|84.85|89.98|79.44|85.00|84.36|
> |CorDA|2e-5|71.56|85.04|76.97|91.58|82.56|85.35|72.01|81.40|80.81|
> |RandLoRA|1e-4|71.96|86.83|79.68|94.64|85.24|89.31|77.05|82.80|83.44|
> |VeRA ($r=512$)|4e-3|62.17|85.15|77.79|91.66|80.27|87.21|75.17|81.60|80.13|
> |IPA|1e-4|74.98|89.93|81.22|96.00|85.87|91.20|79.60|88.40|85.90|
>
> As shown, IPA maintains its advantage over the baselines. We also plan to report LoRA-Null results once the model is fully operational and add the results to the manuscript.
>
> ## Limited theory
> We agree that a formal analysis linking reconstruction error to fine-tuning dynamics and the generalization properties in downstream tasks would be valuable for better understanding _how_ this extra optimization objective contributes to fine-tuning. We consider this an important direction for future work.
>
> ## Limited to linear case
> We limited our study to the linear case to remain within the strict computational budgets typically imposed in the PEFT community for large models, both during fine-tuning and inference. Incremental PCA, built on a randomized low-rank SVD backbone, offers one of the most efficient and theoretically grounded solutions, introducing minimal pretraining overhead and no additional inference cost thanks to the linearity.
>
> Nonlinear extensions, such as variants of the generalized Hebbian algorithm (GHA), which is directly linked to the nonlinear solution suggested by the reviewer, or other forward-only methods, may yield improved reconstruction and fine-tuning performance, but they would also prevent fast inference since the adapter can no longer be directly merged with the pretrained weights.
>
> Given the limited discussion period, we will attempt to include preliminary results in this direction.
>
> ## Generalization and reuse
> We thank the reviewer for raising the question of broader applications of IPA across tasks. The transferability of PEFT modules is an important downstream research direction, connected to topics such as LoRA-based model merging, continual learning, and mixture-of-experts (MoE) models. Recent studies have explored merging or aligning LoRAs through input or output projections shared across tasks, such as Stoica et al. (2025), Tian et al. (2024). We leave a systematic analysis of IPA's transferability for future work.
>
> ## Wall-clock time and memory usage
> For the commonsense benchmark, pretraining the projector on 10\% of the data ($\sim$17k examples) in IPA adds only about 10 minutes of overhead compared to the $\sim$5 hours required for 3 fine-tuning epochs shared by LoRA and IPA. In contrast, DoRA is noticeably slower, requiring around 10 hours under the same configuration. All timings were measured on a single NVIDIA H100 GPU on the same idle node to ensure fair comparison.
>
> During IPA's pretraining stage, memory usage is slightly above the inference budget ($\sim$35 GiB), while fine-tuning memory usage remains identical to LoRA ($\sim$82 GiB while fine-tuning LLaMA3-8B with a batch size of 16). DoRA uses slightly less memory ($\sim$81 GiB).
>
> The small additional pretraining cost comes from using randomized SVD in IPCA, which is substantially more efficient in both time and memory than full SVD. A dedicated paragraph has been added to the manuscript to clarify these details in the ablation study.
>
> References:
> - Stoica et al. (2025), Model merging with SVD to tie the Knots
> - Tian et al. (2024), HydraLoRA: An Asymmetric LoRA Architecture for Efficient Fine-Tuning

---

### Author Response · Authors · 2025-10-14

We thank all reviewers for their feedback and constructive suggestions, which have helped improve both the clarity and completeness of our work.

Below, we summarize the main revisions and additions made in response to their comments:

- **Extended baselines**: We extended our experiments with additional PEFT baselines (PiSSA, OLoRA, CorDA, RandLoRA, and VeRA), using their recommended configurations and ranks to ensure fair comparison.
- **Explicit comparison to random projection**: We added an ablation directly contrasting PCA-based and random orthogonal projections.
- **Wall-clock and memory usage**: We added a dedicated paragraph reporting both the training time and memory footprint. In addition, we included an ablation study comparing IPA with LoRA under an aligned computational budget.
- **Learning rate ablation**: We conducted an ablation across different learning rates for LoRA, DoRA, and IPA to assess robustness.
- **Statistical significance**: We incorporated paired sign tests across tasks (for VTAB results) to evaluate the reliability of the observed improvements.

We have also highlighted the main changes in the updated manuscript. We believe these revisions address all reviewers' concerns and further strengthen the paper.

---

### Decision · Action_Editor_eXhC · 2025-11-18

**Recommendation:** Accept as is

**Audience:**

Yes

**Audience Explanation:**

Yes. Parameter-efficient fine-tuning is an active topic for many TMLR readers who work with large foundation models. This paper highlights a clear and interesting observation about LoRA (which is a widely used approach for this task). Specifically, they show that the down-projection behaves mostly like a fixed projection matrix. The proposed IPA method is simple, practical, and shows consistent improvements over LoRA and DoRA with minimal overhead. Reviewers were positive about this work's contribution and its relevance to the community.


I believe this is a valuable contribution to the community, and I recommend accepting the paper.

**Claims And Evidence:**

Yes

**Claims Explanation:**

The authors tackle the important challenge of parameter-efficient fine-tuning for large models. They propose a method that extends and improves upon the widely used LoRa technique. Their solution is straightforward, logical, and the authors present it effectively.

The paper provides strong empirical evidence to support its main claims. The authors demonstrate that the down-projection in LoRA remains largely unchanged throughout the training process. Furthermore, the study shows that the proposed approach can enhance performance in downstream tasks. The experiments conducted are comprehensive, covering both language and vision tasks, and include various ablations. During the rebuttal phase, the authors presented additional results that illustrate the impact of varying the learning rate and offered a comparison with random projection.